# The Effect of Ethnicity on Identification of Korean American Speech

Andrew Cheng [1,*] and Steve Cho [2]

[1] Department of Linguistics, Simon Fraser University, 8888 University Drive, Burnaby, BC V5A 1S6, Canada
[2] Department of Linguistics, California State University, 18111 Nordhoff Street, Northridge, CA 91330, USA; steve.cho.64@my.csun.edu
* Correspondence: a_cheng@sfu.ca

**Abstract:** Research on ethnic varieties of American English has found that listeners can identify speaker ethnicity from voice alone at above-chance rates. This study aims to extend this research by focusing on the perception of race and ethnicity in the voices of ethnically Korean speakers of English. Bilingual Korean Americans in California provided samples of English speech that were rated by 105 listeners. Listeners rated the speakers on their likelihood of being a certain race or ethnicity (including Asian and White). Listeners who were Korean themselves rated the speakers as more likely to be Asian and Korean, whereas non-Asian listeners rated the speakers as more likely to be White. Non-Asian listeners also demonstrated a negative correlation between rating a voice as Asian and rating a voice as belonging to a native-born American, while Asian listeners did not. Finally, a positive correlation between pitch and perceived Asianness was found for female speakers, corresponding to listeners' metalinguistic commentary about the hallmarks and stereotypes of Asian or Asian American speech. The findings implicate the listener's own ethnic identity and familiarity with a speech variety as an important factor in sociolinguistic perception.

**Keywords:** Korean American; accent; ethnicity; perception; raciolinguistics

## 1. Introduction

The goal of this paper is to demonstrate how a listener's ethnic background can have a powerful effect on their ability to perceive a speaker's ethnic identity. This has been demonstrated previously with American English varieties such as African American English, but has not been done with a specific focus on Korean American English. A secondary goal is to present preliminary evidence for the existence of a variety of English that is associated with Korean Americans, framed around the currently competing approaches to the idea of ethnolectal variation in Asian American communities.

We begin with an introduction to sociolinguistic perception studies centered on race and ethnicity and give a brief overview of the literature on Asian American speech perception. We follow that with a description of the methodology for the main speech perception experiment and the post-hoc perception experiment, which focused on the speech of Korean Californians. The results and discussion sections explain how listener ethnicity, as a proxy for their experience with ethnic varieties of English, interacts with their expectations of what kind of English is normative or "sounds White" in the United States context.

### 1.1. Perception of Ethnic Identity in Speech

Extensive research on ethnic varieties of English in the United States has found that listeners are generally able to identify speakers' race and ethnicity from acoustic properties of the voice alone (Foreman [1998] 2000; Hawkins 1993; Thomas and Reaser 2004). Perhaps the most telling example of this ability and its social consequences comes from Purnell et al. (1999). In this study, listeners were given the task of identifying the ethnicity of a speaker based on voice alone, and the authors reported that listeners were able to

correctly identify both gender and race/ethnicity at a rate of between 79–97%. In the same study, speakers of three broad American English varieties—African American English (AAE), Chicano English (ChE), and Standard American English (SAE)—were used to test housing biases in the San Francisco Bay Area. There was a significant difference in confirmed apartment viewing appointments based on the variety of English spoken that reflected the ethnic population and homeowner rates of each neighborhood, which demonstrates that listeners' ability to distinguish ethnicity through speech may perpetuate existing issues of ethno-geographic stratification and housing inequality. (See also Craft et al. (2020) and Wright (2021), for evidence that this kind of linguistic discrimination persists in the 2020s.)

Further sociolinguistic perception studies have been able to examine more closely which phonetic and phonological characteristics listeners use to determine the ethnic identity of a speaker. In Thomas and Reaser (2004), experimental stimuli were prepared such that stereotypical morphosyntactic and lexical features of AAE, such as copula absence and the invariant habitual *be*, were removed. Listeners were still able to identify the African American voices with greater than 50% selection accuracy, even after various manipulations of prosody and acoustic filtering. Characteristics of both the listener (such as the dialect they speak) and the speaker (including, but not limited to, nonrhoticity, consonant cluster simplification, and prosodic range) affect a listener's successful identification of AAE (Thomas 2007), and it has even been shown that manipulating certain suprasegmental properties of a person's voice can make them "sound blacker" (Holliday and Villarreal 2020).

As for Chicano English, Fought (2002b) describes some aspects of ChE phonology that set it apart from Spanish-accented English, including loss or glottalization of word-final consonants; use of segmental contrasts that non-native speakers do not have; and, at least in the case of Los-Angeles-based Chicano English speakers, a high amount of creaky voice. Whether these cues alone can be used to identify ChE speakers has been less studied, although Fought (2002b) does hypothesize that many of the speech perception and language attitudes studies investigating Spanish-accented speech may in fact be measuring reactions to ChE (Fought 2002b, p. 211). (See also Alfaraz (2014) and Carter et al. (2020) for production and perception studies on Miami English, which has been shown to have ties to Cuban or Caribbean Spanish but, similar to ChE, is much more than simply Spanish-accented English.)

Through these studies, we see that American English speakers as listeners can identify the ethnic identities of speakers of English varieties based on segmental (phonetic) and suprasegmental (prosodic) cues. However, the scope of this research on the perception of ethnolects has historically focused on the perception and identification of AAE and ChE—the two most recognizable and widely spoken ethnolects in the United States context. Meanwhile, we know very little about the perception of speech of various Asian American communities (e.g., Korean Americans), in particular because there is no clearly defined variety of English called "Asian American English" (Reyes and Lo 2009), "Korean American English", or other ethnically-specific varieties.

To be clear, new varieties of English can and do arise, even in the face of modern dialect leveling. This is often seen in the form of new regional dialects, such as California English. The earliest study of California English (Hinton et al. 1987) traced a brief history of its enregisterment as part of the "Valley Girl" stereotype, while also demonstrating how speakers who did not fit that stereotype also used it with increasing frequency. New dialects form, or become enregistered, when a set of signs or linguistic variables becomes associated with a specific group (Agha 2005). Enregisterment of a new variety can occur without influence from other varieties or as a result of language contact, as in the case of Miami English (Carter et al. 2020); it can be more subliminal than overt, as in the case of new Wisconsin English (Schuld et al. 2016); and it can be enregistered with one part of a speaker population but not another, as in the case of Puerto Rican Island English (Font-Santiago 2021). Participation in sound changes associated with a new variety, or even acknowledgment of its existence, can be mediated by all manner of factors related to

social identity, including a speaker's ethnic rootedness (Reed 2018) or ethnic orientation (Hoffman and Walker 2010). Following the trends in these investigations of new varieties of American English (ethnic or otherwise), we turn now to a discussion of Asian American ethnic identity formation.

### 1.2. Asian American Identity and Speech Perception

The United States government currently identifies "Asian" as a racial category, equivalent in the legal sense to "White" and "Black/African American". "Hispanic/Latino" is an ethnic category that is not mutually exclusive to any racial category[1]. The category of "Asian" is further broken down into ethnic subcategories that pertain to countries of origin[2] for Asian people, who are often (but not always) immigrants and descendants of immigrants. Therefore, "Korean" and "Korean American" are ethnic categories that circulate in discussions of group identity, while "Asian" and "Asian American" are racial categories. Yet, it is important to note that these terms, most of which were coined following the civil rights movements in the 1970s (Espiritu 1992), are conceptualized differently among the varying communities of practice that use them, including academic research.

An early study of the perception of Asian American speech showed evidence for a perceptible variety. Hanna (1997) used natural speech samples from native-born Americans of Asian (including Chinese, Korean, and Filipino) and Caucasian background who grew up in the Philadelphia metropolitan area. Listeners, who were told that the samples they heard all came from local native speakers of English, were given a forced-choice identification task (White or Asian). Hanna reported that regardless of their own ethnicity, listeners were able to identify the Asian Americans at above chance rates.

A more recent study, Newman and Wu (2011), sampled speakers and listeners who were raised in New York City and its surrounding suburbs. There were two African American speakers, two Latinx ChE speakers, two European Americans, four Korean American speakers, and four Chinese American speakers, all of whom had acquired English natively. Listeners were able to identify Asian American speakers at a rate significantly higher than chance. Second generation Asian American listeners were slightly more successful than non-Asian listeners at identifying other Asian American speakers.

Finally, Wong and Babel (2017) conducted a perceptual study of Chinese, East Indian, and White Canadians raised in the Metro Vancouver region of Canada. Listeners heard two-second clips of spontaneous speech from these speakers and were given a forced-choice identification task (White, Chinese, or East Indian). In addition, listeners were asked to rank their amount of association with each of these three ethnic groups in their own social networks. Findings indicated that listeners of each network correctly identified the speakers' ethnicity, with White Canadian speakers being identified with the highest accuracy, followed by Chinese Canadian speakers, but also that a listener's own network influenced their accuracy.

Each of these studies, while focused on the perception of speech, also took a closer look at the phonetic production of the voices that were most frequently identified as sounding Asian. Hanna (1997) hypothesized that a high-rising pitch contour throughout an utterance is characteristic of Asian American voices, while Newman and Wu (2011) analyzed several potential acoustic cues, including breathy phonation, Voice Onset Time (VOT), and formant quality of prevocalic /ɹ/, which all showed up in various combinations in the Asian American speech samples. Wong and Babel (2017) sampled Asian Canadian speakers, rather than Asian American speakers, but found some similar acoustic properties that were potentially attributable to a transfer effect from Cantonese into (North) American English, including a less-constricted postvocalic /ɹ/, /ð/-stopping (e.g., *this* pronounced as *dis*), and less reduction in unstressed syllables.

However, consensus on what acoustic cues the Asian American listeners are focusing on when they successfully identify a fellow Asian American speaker has not yet been reached. Variationist research on the production of English sociophonetic variables—most often variables undergoing sound change—in local Asian American speech communities is

growing (Bauman 2016; Ito 2010; Jeon 2017; Wong and Hall-Lew 2014). However, singular variables alone are not enough to identify a speaker as Asian American, since one variable has a host of potential social meanings in its indexical field (Eckert 2008), and no one speaker necessarily makes use of all of the variables at all times (Sharma 2011). Only in combination with many other (linguistic, auditory, and sometimes visual) cues can a speaker provide the listener enough information to make an accurate identification. Due to a dearth of research, the full set of phonetic variables that constitute either Asian American English, broadly defined, or a specific Korean American English, has not been determined.

When Reyes and Lo (2009) wrote that there was insufficient evidence to claim the existence of "Asian American English", they had in mind the truth that Asian American speakers are very diverse, and their speech styles and other linguistic behaviors were drawn from a variety of sources, not one hegemonic Asian American identity. This critique is part of the growing contest surrounding the term "ethnolect" and the social and theoretical problems that may arise if linguists uncritically assume that ethnic varieties must exist because ethnic categories do (Jaspers 2008). Yet, evidence from the literature continues to mount that some Asian Americans may indeed sound distinct; they do not unilaterally speak "standard" or "mainstream" United States English (which, given the history of variationist sociolinguistic inquiry, is usually implied to be the speech of "native-speaking" White non-Hispanic Americans, usually educated and middle-class).

As has been shown with perceptions of AAE (Rickford and King 2016) and many other language varieties (Dragojevic et al. 2018; Nagy et al. 2020, inter alia), the impact of "hearing an accent" in someone's voice is real and significant. Student perceptions of Asian-accented instructors weighs heavily on their perceptions of the quality of the instructor (Rubin 1992; Rubin and Smith 1990). In some cases, if listeners perceive an accent to be "foreign," they also perceive the speaker to be a poor teacher. It is possible that there are similar ramifications of speaking an identifiably Asian American English, due to implicit associations of Asian identity with foreignness (Cheryan and Monin 2005; Yogeeswaran and Dasgupta 2010).

One historic example is the case of Judge Lance Ito, a third generation Japanese American. During the highly publicized 1995 O.J. Simpson trial in Los Angeles, he was subjected to mockery for his "Asian accent", despite speaking Standard American English (Baugh 2003, p. 166). In these ways and more, Asian Americans have been marginalized and discriminated against in American society, misrepresented as "forever foreigners" (Tuan 1998) who carry the xenophobic stigma of being a foreigner regardless of their phenotype, citizenship status, country of origin, or command of English.

### 1.3. Korean Americans and English

On top of the relative dearth of sociolinguistic studies on the English of Asian Americans as a whole, there is very little published research to date that focuses specifically on the speech of Korean Americans and its perception or identification by listeners who are either familiar or unfamiliar with Korean and Korean American culture.

Korean Americans are a demographic of Americans whose ethnic heritage is from the Korean peninsula, including present-day North and South Korea. Korean Americans have lived in the United States and its territories since the early 1900s, but larger waves of immigration began after the Korean War in the 1950s and reversal of the Asian Exclusion Act in 1965. Today, over 1.7 million Korean Americans reside in the United States, of which approximately 450,000 were born in the United States. These are considered "second generation" Korean Americans, while their parents who immigrated as adults are "first generation". Second generation Korean Americans are likely to grow up bilingual in Korean and English, but acquire English with native proficiency. A third immigration category is "1.5 generation" Korean Americans (Park 1999), who are born in Korea but immigrate as children with their families and experience significant portions of their childhoods in both countries, immersed in both languages.

Young Korean Americans are important to study from the perspective of understanding Asian American speech, because while Korean Americans are a small minority even within the minority demographic of Asian Americans, they have an outsize cultural influence, especially in large urban centers that influence culture on a national scale, such as the Los Angeles metropolitan area. Southern California is not only well known for the culture of Hollywood and the cultural and linguistic stereotype of the Valley Girl (D'Onofrio 2015; Fought 2002a; Hinton et al. 1987) that emerged in the 1980s and continues to expand today (Podesva 2011), but also boasts one of the highest populations of Koreans outside of Korea, many of them based in the diverse neighborhood in midtown LA called Koreatown (Sanchez et al. 2012).

Korean Americans who grew up in and around enclaves such as Koreatown have developed a blended cultural identity (Park and Kim 2008) that has plausible linguistic consequences. However, while there is no shortage of research in cultural psychology pertaining to the ethnocultural practices and beliefs of Korean Americans and the effects of assimilation, biculturalism, and the like, the main focus of linguistic inquiry has tended to be on the maintenance of Korean as a heritage language in this populations (Lee 2002; Shin 2005, 2016) rather than on the characteristics of Korean Americans' English. Despite the overall under-representation of Korean American English in sociophonetic literature, three recent studies examined the production of some segmental and suprasegmental variables, including the THOUGHT and TRAP vowels of Koreans in New Jersey (Lee 2016), the GOAT vowel of Koreans in Texas (Jeon 2017), and the vocalic fundamental frequency of Koreans in California (Cheng 2020a). These studies are limited to production work and do not address the perception of Korean American speech.

Whether in the context of Koreatown or the larger scope of Korean American identity and community, listeners in the "in-group", who have more experience hearing the voices of Koreans and Korean Americans, may have stronger associations between the acoustic signals that Koreans tend to produce and the social category of "ethnic Korean". On the other hand, listeners who are not Korean, who may have not been exposed to much Korean American speech, may categorize the voice as having an unfamiliar accent, or perhaps no accent at all[3]. In the context of the United States, where White speakers are a majority, having "no accent" can be equated in many minds as being a White, native-born American (Devos and Banaji 2005), and any percept of a non-normative voice might be attributed to a speaker having been born and raised in a non-English-speaking place. However, Korean Americans with strong associations between Korean American voices and an American-born identity would be less likely to associate perceptible accentedness with being foreign-born.

In pursuit of a better understanding of the perception of English from minoritized speech communities, we seek to test if listeners are able to identify English-speaking voices of Korean Americans as belonging to Koreans, and whether Korean American listeners and non-Korean American listeners will differ in their ability.

Our hypotheses are as follows:

**Hypothesis 1 (H1).** *Without prior knowledge of the speaker, listeners will correctly identify certain English-speaking voices as sounding distinctly Korean American. (Null hypothesis: Listeners will never be able to identify the voices as Korean American at above chance.)*

**Hypothesis 1a (H1a).** *The ethnicity of the listener will influence the abovementioned ratings. Listener familiarity with Korean American culture may also have an effect on ratings.*

**Hypothesis 2 (H2).** *Identification as "sounding Korean American" will correlate strongly but not perfectly with sounding "non-native" (again, with the ethnicity of the rater affecting this relationship). (Null hypothesis: listeners will identify speakers as sounding Korean American, non-native, or neither, but never a mix of both.)*

## 2. Materials and Methods

Samples of natural speech were taken from thirty-nine Korean Americans with two parents who were ethnic Koreans. All of the speakers were either born and raised in the United States or had immigrated to the United States prior to the age of 16. In terms of generational status, twenty-six were categorized as second generation and thirteen as 1.5 generation, including four early arrivals (immigrated between the ages of 3 and 8) and nine late arrivals (immigrated between the ages of 8 and 16). All of the speakers self-reported as bilingual in Korean and English and were either dominant in English or balanced bilinguals (equally proficient in both languages). They were recorded during a casual interview about their lives, hobbies, and perspectives on language and culture that was part of a larger study. The speakers gave informed consent for their speech to be recorded and analyzed and were compensated monetarily for their participation.

Audio was recorded in one of two ways: either using lapel microphones in a sound-proofed recording studio, or using a Zoom H4N portable recorder in an indoor space such as an office or a library study space. Audio was sampled at either 44.1 kHz or 48 kHz, and at 16 or 32 bits per sample[4].

From the interview audio of each interviewee, two short excerpts of speech were extracted to create a short audio clip in mono. Each clip was between 1 and 3 s in duration. Special care was taken to ensure that the acoustic quality of the clip was good (i.e., free from overlapping speech, nonspeech sounds, and background noise) and that there was no semantic content in the speech that would point to the speaker's identity (e.g., mentions of race, ethnicity, age, gender, sexuality, and geographic location). Each clip was also extracted at word boundaries such that no word was ever cut off, but not necessarily at prosodic or syntactic boundaries. Transcriptions of all the sentences used in the study can be found in Appendix A.

The perception study was run on Qualtrics, and 107 listeners were recruited through a combination of Amazon Mechanical Turk and the authors' personal networks. For the perception portion of the experiment, listeners used personal audio devices to listen to the audio clips: approximately one-third used earbuds or in-ear headphones; one-third used over-ear headphones; and one-third used external speakers. (Two listeners were excluded due to having used speakers on a mobile device, which were considered too low quality.) Out of the resulting 105 listeners, 19 identified as being ethnically Korean, 18 identified as Asian but not Korean, 44 identified as White/Caucasian only, and 24 identified as other races and ethnicities, including Black, Hispanic/Latino, Other, and mixed race (but not Asian). In total, there were 37 Asian listeners and 68 non-Asian listeners. All listeners were located in the United States, gave their consent to participate in the study, reported no history of hearing problems, and were compensated monetarily for their participation.

The study began with two experimental listening blocks: A and B. In block A, listeners heard thirty-nine audio clips and rated the speaker they heard in the clip on a five-point Likert scale of the speaker's perceived likelihood of being any of the following races/ethnicities: "White/Caucasian, "Black/African American", "Asian American", and "Hispanic or Latino" (Figure 1). If a listener rated a speaker as likely to be Asian American, they were given the opportunity to further specify what kind of Asian ethnicity, choosing as many options as they liked from the following list: Korean American, Chinese/Taiwanese American, Vietnamese American, Japanese American, and Other (fill in the blank), as in Figure 2. The opportunity to specify a more detailed ethnicity was not given for choosing White, Black, or Hispanic/Latino[5].

In block B, listeners heard thirty-nine different audio clips from the same speakers (as there were two clips for each speaker) and rated the speaker they heard in the clip on several Likert scales of the speaker's demographic and personality traits, adapted from Campbell-Kibler (2007), including their perceived shyness, perceived friendliness, perceived level of education, perceived geographic place of origin (urban or rural) and perceived likelihood of being American-born and raised (versus foreign-born). This last item was important to include in order to differentiate between listeners who associated

Asian ethnicity with foreign-born status and those who did not; all of the others were intended as distractors.

Listeners were informed that the study was designed "to examine the social valuation of English speech". As the listeners were not given any identifying information about the speakers before they began, they did not know that all thirty-nine speakers were, in fact, Korean American.

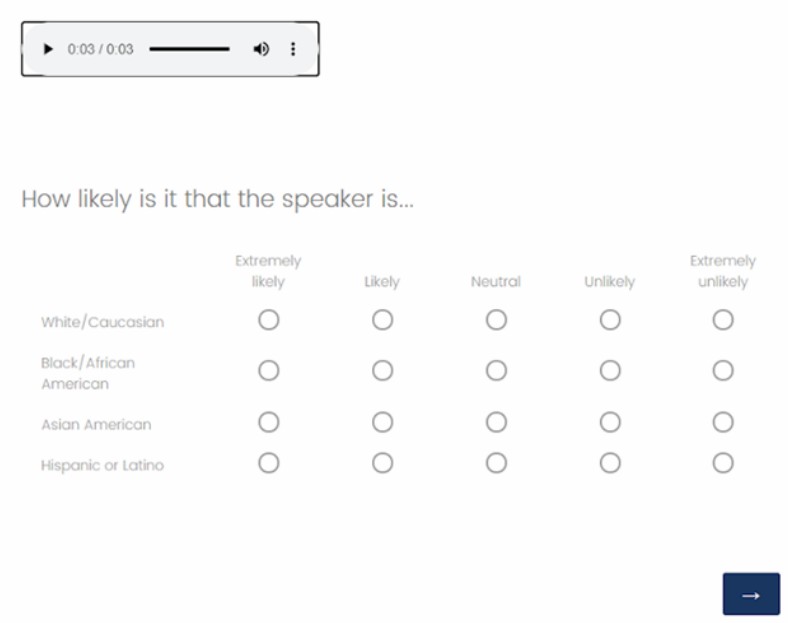

**Figure 1.** An example of the first race/ethnicity rating Likert scale in block A.

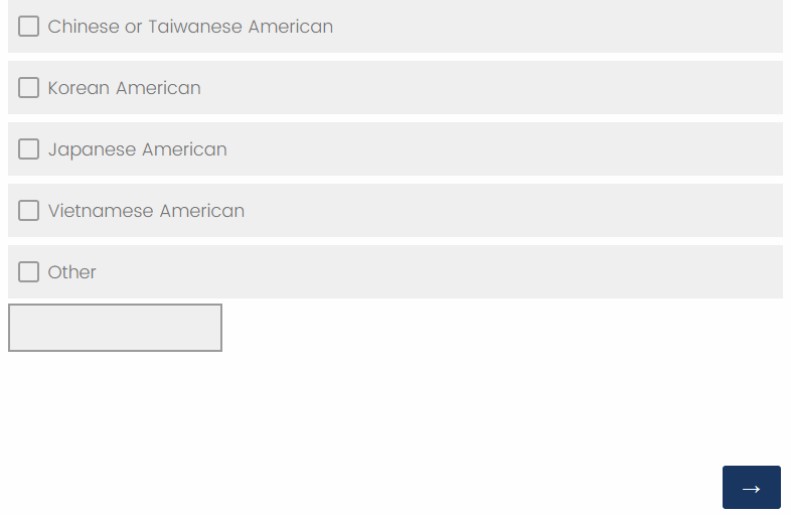

**Figure 2.** If "Extremely likely" or "likely" were selected for "Asian American", listeners were given a follow-up question to further specify what kind of Asian.

Blocks A and B were themselves randomly presented to the listeners. After completing both blocks, listeners filled out a language background questionnaire that included questions about their ethnic background, age, immigration status, and familiarity with the Korean language, as well as one question that asked for their opinions on the characteristics of "Asian American speech". This was intended to elicit metalinguistic commentary or folk linguistic narratives (Albury 2017), which could then be cross-referenced with the experiment results.

In addition to the main speech perception study, we conducted a post-hoc control study of the semantic content of the speech clips to ensure that listeners would not be able to determine the speaker's race based on lexical, syntactic, semantic, or pragmatic information provided by the words spoken. To do this, we ran a separate study that was similar in all aspects to the main study, except that we replaced the audio clips with written transcriptions of the content of each clip. Twenty participants were asked to silently read the transcriptions and then make the same judgments about speaker ethnicity and background, without any acoustic information to base their judgment on. Our expectation was that the participants would essentially rate all the "speakers" as neutral for every social category (i.e., neither likely nor unlikely to be White, Asian, foreign-born, etc.).

## 3. Results

### 3.1. Perception of Race/Ethnicity

The perception of race/ethnicity was measured by converting the Likert-scale responses to a numeric response, with "Extremely unlikely" corresponding to "1" and "Extremely likely" corresponding to "5". Listeners were allowed to judge speakers as being likely or unlikely to be any of the four possible categories. Thus, a higher average score for one race/ethnic category for a particular speaker does not entail lower scores for the other categories. A mean rating was calculated for each speaker for each category, with a score of 3 indicating neutrality (i.e., on average, listeners determined the voice to be neither likely nor unlikely to be of a certain category).

First, we found that two demographic characteristics of the speaker had no effect on their ethnicity ratings. The generational status of the speaker (being 1.5 generation or second generation[6]) did not affect their score on the "likelihood of being Asian" scale (henceforth, "perceived Asianness" or "mean Asian rating"). A one-way ANOVA showed no statistically significant difference between generational groups ($F(1,154) = 0.193$, $p = 0.66$). Generational status also did not significantly affect mean ratings for any other ethnicity or for American-born status.

The second characteristic was Age of Arrival (AOA), which was also found not to affect ratings of perceived Asianness. A simple linear regression model was fit to the mean Asian rating data and speakers' AOA. The mean Asian rating could not be predicted from AOA (adjusted $R^2 = 0.01$, $p = 0.2$). However, weak correlations were found between AOA and perceived Whiteness ($R^2 = 0.04$, $p = 0.03$) and perceived American-born status ($R^2 = 0.03$, $p = 0.04$). Figure 3 illustrates downward trends for the lines, indicating perceived Whiteness and Americanness, whereas the lines indicating perceived Asian (as well as Black and Hispanic/Latino) identity are relatively flat and have wide confidence intervals.

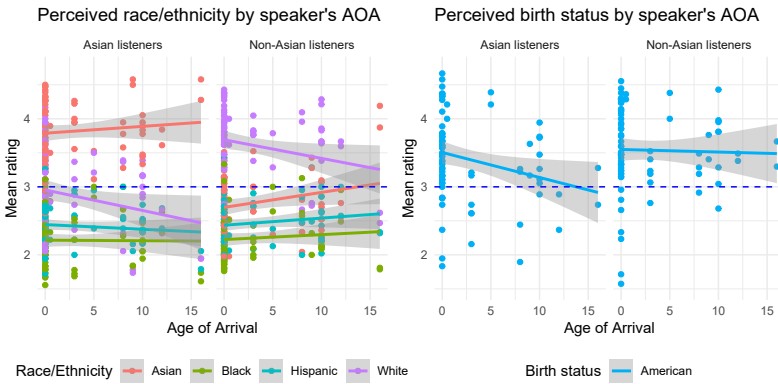

**Figure 3.** Age of Arrival was shown to have a modest effect on mean White ratings and mean American ratings, but not on perceived Asian, Black, or Hispanic/Latino identity. In addition, Asian listeners (n = 37) and non-Asian listeners (n = 68) rated the speakers very differently in terms of perceived Whiteness and Asianness.

Regardless of the slopes of the lines, however, the figure also shows a distinct difference in the intercepts of the lines depending on the listener's ethnic identity. This is illustrated explicitly in Figure 4, which shows the distributions of ratings for each racial/ethnic category and American-born status. Overall, Asian listeners gave higher Asian ratings and lower White ratings compared to non-Asian listeners, but both groups of listeners gave similar ratings for likelihood of being American-born.

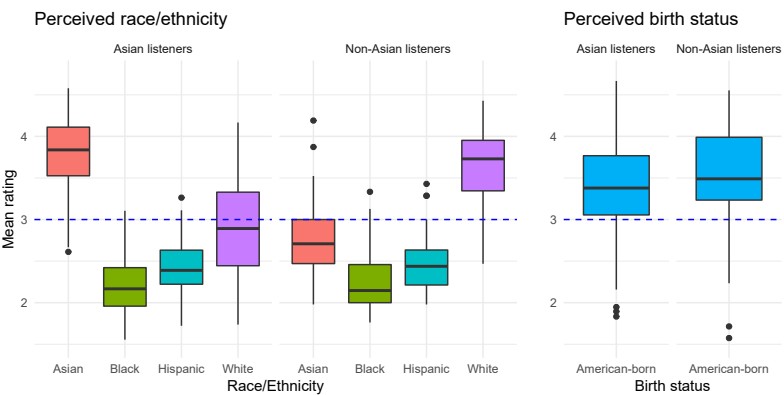

**Figure 4.** Asian listeners (n = 37) were more likely to rate the Korean American voices as being Asian but not White, whereas non-Asian listeners (n = 68) showed the opposite tendency, identifying most of the Korean American voices as likely to be White, but not Asian. Both listener groups identified the speakers as being, on average, more likely to be American-born than foreign-born.

This leads us to an analysis of the relationship between mean Asian and White ratings among listeners based on listener ethnicity: Korean, non-Korean Asian, non-Asian Person of Color (POC[7]), and White only.

Results showed that regardless of listener ethnicity, there was a strong negative correlation between ratings for perceived Asianness and perceived Whiteness. That is to say, the more likely a speaker was judged as being White, the less likely they were judged as being Asian. The scatterplots in Figure 5 show this downward trend for all four groups: Korean, non-Korean Asian, non-Asian POC, and non-POC (White). The same thirty-nine speakers are shown in each plot, but with different coordinates based on the average scores of each listener group.

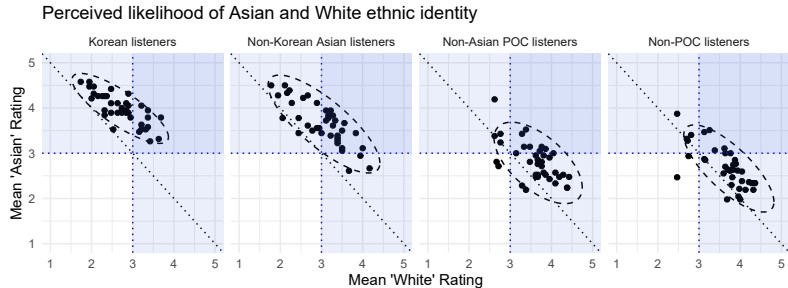

**Figure 5.** All four listener groups (Korean n = 19, non-Korean Asian n = 18, non-Asian POC n = 24, White n = 44) demonstrated a robust negative correlation between perceived Whiteness and perceived Asianness. Asian listeners were more likely to perceive the speakers as Asian, while non-Asian listeners (whether POC or White) were more likely to perceive the speakers as White.

This correlation was confirmed using a Kendall correlation test. This test was chosen because of a skew toward perceived Whiteness among the White listener group. The test showed a statistically significant negative correlation between mean Asian rating and mean White rating for Korean listeners (Kendall's tau = −0.6, $p < 0.001$), non-Korean Asian listeners (Kendall's tau = −0.62, $p < 0.001$), non-Asian POC listeners (Kendall's tau = −0.4, $p < 0.001$), and White listeners (Kendall's tau = −0.54, $p < 0.001$).

As discussed earlier, the relative mean Asian and White ratings for each listener group were also different. Figure 5 uses dashed lines at the "3" point on each axis to represent the "neutral likelihood" or "unsure" choice on the Likert scales; a mean score greater than "3" represents a speaker who was judged likely to be a certain ethnicity, and less than "3" represents a speaker who was judged unlikely to be a certain ethnicity. Among Korean listeners, all thirty-nine speakers were judged above "3" in mean Asian rating, but centered under "3" in mean White rating. Among non-Korean Asian listeners, most speakers were judged above "3" in mean Asian rating, but centered at "3" in mean White rating. The ratings given by non-Asian listeners were greatly skewed toward perceived Whiteness, with both non-Asian groups' ratings of Asianness centering below "3" and ratings of Whiteness centered around "4". There also appears to be greater dispersion of ratings among all non-Korean groups compared with the Korean listeners.

A one-way ANOVA test showed a significant effect of ethnicity group on the mean Asian rating ($F(3,152) = 85.14$, $p < 0.001$). A Tukey HSD post-hoc test showed that all four groups differed significantly from one another in mean Asian rating, with the exception of the White listener group and the non-Asian POC listener group, who gave similar mean Asian ratings.

A second ANOVA test showed a similar significant effect of ethnicity group on the mean White rating ($F(3,152) = 27.3$, $p < 0.001$). The post-hoc test showed that for perceived Whiteness, the two Asian listener groups did not differ significantly in scores, nor did the two non-Asian listener groups, but every other group comparison was significantly different.

### 3.2. Perception of Race and Foreign-Born Status

Listeners also judged the same thirty-nine speakers on a range of personal characteristics besides race, including their perception of a speaker as having been born in America (i.e., the United States) or in a foreign country. Once again, all listener groups showed a similar negative association, with speakers judged as being more likely to be American-born also judged as less likely to be Asian. However, this association was not as strong across all groups, compared with the association between Asian and White ratings. Statistical tests showed a statistically significant negative correlation between mean Asian rating and mean American-born rating for Korean listeners (Kendall's tau = −0.27, $p = 0.02$), non-Korean Asian listeners (Kendall's tau = −0.29, $p = 0.01$), non-Asian POC listeners (Kendall's tau = −0.38, $p = 0.001$), and White listeners (Kendall's tau = −0.33, $p = 0.004$). The strength of the association was slightly greater for non-Asian listener groups than Asian listener groups (Figure 6).

The positive association between being American-born (versus foreign-born) and being White was even stronger than the negative association between being American-born and being Asian, and it was equally strong across all four ethnic groups.

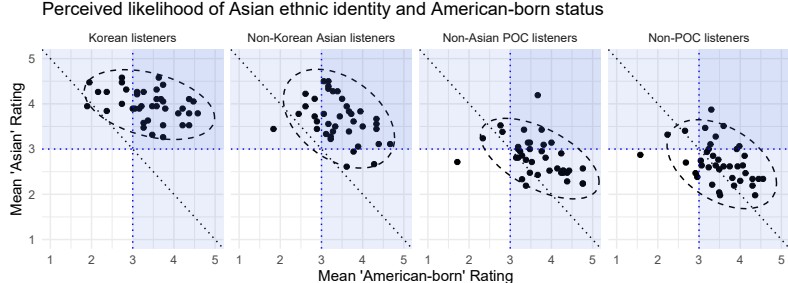

**Figure 6.** All four listener groups (Korean n = 19, non-Korean Asian n = 18, non-Asian POC n = 24, White n = 44) demonstrated a weak negative correlation between perceived American-born status and perceived Asianness. The correlation was weaker among Asian listeners than among non-Asian listeners.

The relative accuracy of listeners or listener groups in terms of their perception of American-born status was not calculated, as it was already determined that generational status (which is determined by birthplace among second generation and 1.5 generation Korean Americans) did not affect the perception ratings of speakers.

### 3.3. Perception of Specific Asian Ethnicity

Finally, an analysis of the specific Asian ethnic categories chosen for each of the speakers revealed further differences between listener groups. In this analysis, the ratio of the number of each Asian ethnic category choice to the total possible choices was compared with the rating of likelihood of being Asian. For example, if Speaker 1 was perceived as "not likely" to be Asian by half of the listeners, then half of their specific Asian ethnic category responses would be "None" (as the opportunity to specify was not given to them). If, among the listeners who selected "likely", the box for 'Korean American" was selected 50 times, "Chinese or Taiwanese American" was selected 25 times, "Vietnamese American" was selected 25 times, and Japanese and Other were not selected, then the ratio for Speaker 1 would be 0.5 Korean American, 0.25 Chinese or Taiwanese American, and 0.25 Vietnamese American. The raw count of each specific Asian ethnic category was not considered, because a listener was free to choose multiple Asian categories; this means that the sum total of selections exceeded the number of listener participants, and the total of ratios for any particular speaker may exceed 1. Thus, accuracy was judged based on the value of the Korean ratio: the higher this number, the more respondents in each particular group identified the voice as Korean relative to the other options.

Results showed that among all four listener groups, the ratio of "None" responses dropped considerably as the mean Asian rating increased, which is a logical consequence of listeners being prompted to select a specific Asian ethnic category when they selected "likely" or "extremely likely" on the Asian scale. This is shown in Figure 7 as the strong negative-sloping line in each plot.

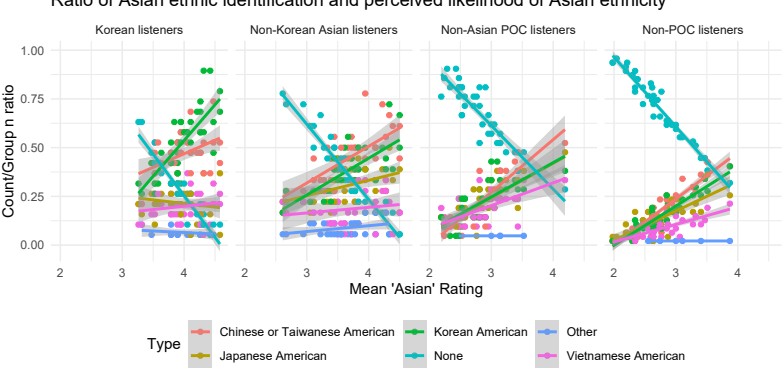

**Figure 7.** Ratio of the number of times each ethnicity was chosen to the total number of choices made for each speaker, by speaker's mean Asian rating, split by listener's racial/ethnic category (Korean n = 19, non-Korean Asian n = 18, non-Asian POC n = 24, White n = 44). Korean listeners were most likely to identify Asian American speakers as specifically Korean American, while the other three listener groups tended to choose Korean and Chinese or Taiwanese with equal frequency.

However, among non-Asian POC listeners and White listeners, the ratio of "None" responses was around 0.25 even at the highest mean Asian rating (on the right edge of each plot), while it was near 0 for the two Asian listener groups. This indicates that after identifying a voice as likely to be Asian, Asian listeners almost always chose a specific Asian ethnic category, while non-Asian listeners only did so 75% of the time.

It is also apparent from Figure 7 the specific Asian ethnic category that was most often selected for each listener group, and how the rate of selection changed as the mean Asian rating increased. Korean listeners (n = 19) identified most speakers as Korean American, and the higher the mean Asian rating, the higher the ratio became. Chinese or

Taiwanese American was the second most likely choice, while Japanese and Vietnamese were unlikely choices. Among the 18 non-Korean Asian listeners, Korean American and Chinese or Taiwanese American were equally likely choices when a speaker was perceived as very likely to be Asian, with Chinese or Taiwanese American slightly edging out Korean American.

A further analysis of the non-Korean Asian listeners revealed that Chinese and Taiwanese listeners (n = 11) were slightly more likely to select Chinese or Taiwanese American as the specific Asian ethnic category, but Vietnamese listeners (n = 3) were not more likely to select Vietnamese and, in fact, most often made no selections at all (Figure 8). Thus, the pattern for non-Korean Asians seen in Figure 7 was driven primarily by the Chinese and Taiwanese listeners in this group.

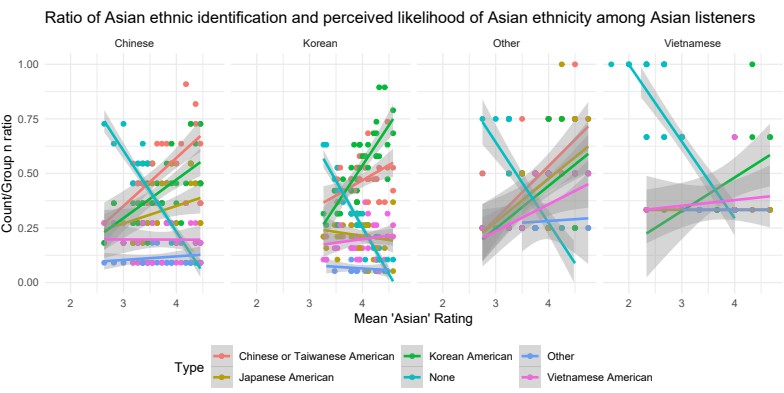

**Figure 8.** Ratio of the number of times each ethnic category was chosen to the total number of choices made, for Asian listeners only, split by listener ethnic group. Korean listeners (n = 19) were most likely to identify Asian American speakers as specifically Korean American, while Chinese and Taiwanese listeners (n = 11) were more likely to select Chinese or Taiwanese. Vietnamese listeners (n = 3) were not more likely to select Vietnamese, but made no selection at all more often than the other Asian groups. Four listeners in the "Other" category behaved similarly to the Chinese and Taiwanese listeners.

However, among non-Asian listeners, all four Asian ethnic categories were equally likely to be selected, and no single category particularly stood out from the others. Another way of putting it is that since all of the speakers were Korean American, it was the Korean listener group that was the most accurate in its identification of the ethnic category of the speakers, while the three non-Korean groups performed no better than chance (i.e., Korean American selections made up 50% of total ethnic category selections or less).

*3.4. Post-Hoc Test of Text Stimuli*

In the post-hoc test of the stimuli, we presented raters with only a text transcription of the speech from the main experiment, to make sure that no lexical, syntactic, semantic, or pragmatic information in the audio samples had provided listeners with information about the ethnic identity of the speaker. We expected the raters not to be able to tell a speaker's ethnicity from the transcribed speech, and that the responses would cluster around "3", or neutral, for every speaker.

Instead, among non-Korean, non-Asian listeners, the ratings for perceived Asianness still clustered below "3" and the ratings for perceived Whiteness were above "3". In other words, the raters perceived the speakers of the text that they read to be White and not Asian. Only one participant in the post-hoc study was of Korean descent, and this participant rated every text as "3" all the way down. These results are shown in Figure 9.

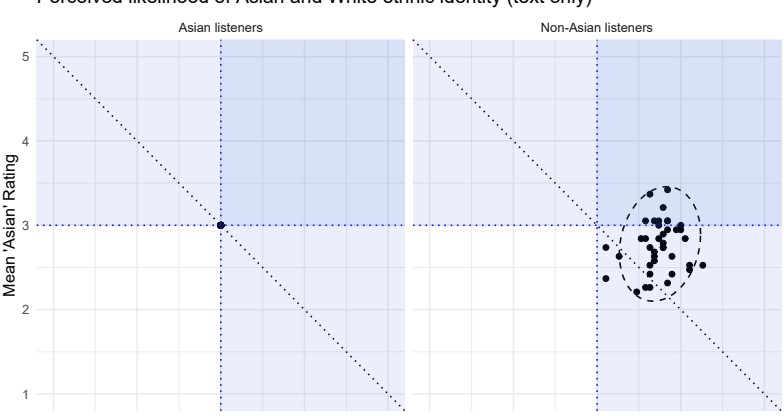

**Figure 9.** Without audio, and only given text to read, the non-Asian post-hoc experiment participants still rated most speakers as likely to be White and not likely to be Asian. One participant in the post-hoc study was Korean and rated almost every speaker as neutral.

*3.5. Listener Metalinguistic Commentary and Acoustic Case Studies*

Listeners were asked at the end of the experiment to provide open-ended commentary on what they believed an Asian American voice sounds like. The text of the question was as follows: "In your own words, what aspects of a person's voice might characterize them as having an 'Asian American' accent? (Or specifically 'Korean American', 'Chinese American', etc.) Be as specific or as general as you wish".

Korean American respondents, whether they had been born and raised in California or not, tended to mention suprasegmental attributes of speech such as tone and prosody. For example, mentions of Asian Americans having a higher pitch and a certain 'cadence' or 'intonation' that set them apart occurred thirteen times (out of twenty respondents). Korean Americans also mentioned that Asian Americans might have a softer tone, a lower voice, and vocal fry (i.e., creaky voice). A few mentions were made of specific segmental features: /ð/-stopping, /ɪ/ as [i] (*bit* as *beet*), and /oʊ/ (the vowel in *goat*) sounding "rounder than it does for white people". Interestingly, several Korean American listeners who had been raised both in and outside of California mentioned the Valley Girl stereotype, in conjunction with discussion of the "intonation" of Asian Americans ("ending a statement in a question"), as well as "high use" of the filler word *like*.

Among non-Korean listeners, there was a high amount of variation in responses. Nevertheless, some themes rose to the surface, only one of which had a counterpart in the Korean American listeners' responses. Out of 86 responses, mentions of a higher pitch among Asian Americans and different tones, inflections, or cadences occurred twenty times. Other common responses included that Asian Americans had softer, quieter, or lighter speech, pronounced specific words incorrectly or differently, pronounced vowels differently (usually longer or more drawn out), and also tended to have "clipped" words and syllables.

We performed both an impressionistic aural analysis and a basic acoustic analysis of the speech that was rated most likely to be White, Asian, and American-born among all listener groups. Some acoustic characteristics of the speech that was rated most likely to be Asian include /ð/-stopping, a relatively long VOT or aspiration in word-initial and word-medial /t/, and word-final devoicing. Among speakers who were rated as most likely to be American-born, we observed relatively faster speech rate and raised pitch at prosodic boundaries (also called high rising terminals, or "uptalk"). These were particularly apparent in two speakers (08 and 30) who were rated as particularly likely to be both Asian and American-born by the Korean listeners.

**Speaker 08 (female):**

8A. Yeah I thought it wasˆ (.2) pretty okay u:m **(0:02)**

*Asian Ethnicity rating: 4.42 (Korean listeners); 3.00 (White listeners)*

*White Ethnicity rating: 2.47 (Korean listeners); 3.64 (White listeners)*

8C. It's hard to: identify and like- (.6) see similarities **(0:04)**

*American-born rating: 3.74 (Korean listeners); 3.91 (White listeners)*

**Speaker 30 (female):**

30A. You knowˆ (.) I'd say it comes back (.3) fairly quickly **(0:03)**

*Asian Ethnicity rating: 4.11 (Korean listeners); 2.64 (White listeners)*

*White Ethnicity rating: 2.63 (Korean listeners); 3.70 (White listeners)*

30C. I think that no::w they care a lot because **(0:03)**

*American-born rating: 4.37 (Korean listeners); 4.06 (White listeners)*

For these two female speakers, we hypothesize that suprasegmental properties of their voices, including voice quality, speech rate, and pitch raising at prosodic boundaries (indicated by the carets), carried the most information about their ethnic identity to listeners. Female speakers did, in fact, have a stronger correlation between mean pitch (f0) and their ethnic identifications by all listeners, as the acoustic analysis confirmed.

Our basic acoustic analysis of the speech included mean pitch and pitch range of the voice samples used in the ethnic identification task. Fundamental frequency (f0) in Hertz was extracted from the 39 samples using Praat (Boersma and Weenink 2021). Each speaker's mean pitch was plotted against their mean ratings for Asian or White identity (Figure 10). We then ran Kendall correlation tests on the relationship between mean pitch and mean Asian or mean White rating, and between pitch range and each ethnicity rating, split by speaker gender and listener race. Selected results are displayed in Table 1.

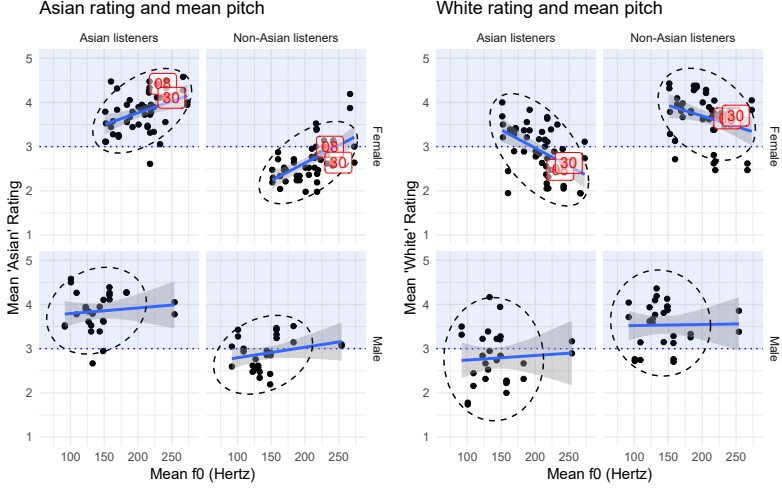

**Figure 10.** Among both Asian and non-Asian listeners, a higher pitch (fundamental frequency, or f0) correlated with perceived Asianness for female speakers, but not for male speakers. Two speakers with high Asian ratings and relatively high-pitched voices are labeled and discussed in Section 3.5.

Indeed, the speakers who were rated as most likely to be White also had the lowest-pitched voices, and the speakers who were rated as most likely to be Asian had the highest-pitched voices; however, this was only true for female speakers. There were no significant correlations between vocal pitch properties and ethnicity ratings for the male speakers. As for pitch range, there was a significant correlation between a larger pitch range and higher Asian ratings among non-Asian listeners, but not among Asian listeners (given an alpha criterion of 0.01), and not for ratings of Whiteness.

**Table 1.** Correlation test results ($\tau$ = Kendall's tau) for ethnic category ratings by vocal pitch properties, split by speaker gender and listener racial/ethnic group. For female speakers, mean pitch (f0) correlated strongly with identification as both Asian and White, but pitch range did not. Male speakers demonstrated no correlation between vocal pitch properties and ethnicity ratings.

| Speaker Gender | Speaker ID | Listener ID | Corr. (Mean f0) | Corr. (f0 Range) |
|---|---|---|---|---|
| Female | Asian rating | Asian | $\tau = 0.30$, $p = 0.002$ | $\tau = 0.21$, $p = 0.034$ |
| | | Not Asian | $\tau = 0.41$, $p < 0.001$ | $\tau = 0.31$, $p = 0.002$ |
| | White rating | Asian | $\tau = -0.37$, $p < 0.001$ | $\tau = -0.08$, $p = 0.42$ |
| | | Not Asian | $\tau = -0.27$, $p = 0.007$ | $\tau = -0.002$, $p = 0.98$ |

Thus, we conclude that the listeners' metalinguistic commentary corresponded with their perceptual behavior: there was a tendency for listeners to equate "Asian-sounding" voices with higher-pitched voices, and the voices that tended to be rated as more likely to be Asian had higher fundamental frequency. Note that we cannot claim that female Asian Americans do indeed have higher-pitched voices compared with female speakers of other races, since every speaker in this sample was Asian American. Rather, the perception of Asian identity in female voices does appear to be closely tied to pitch and other suprasegmental vocal properties. However, we acknowledge that there were many other acoustic variables present in the signal (such as vowel quality or voice quality) that listeners may have attended to during the task, consciously or not, and none of these were explicitly controlled for during the study.

In the end, listeners conceded that it was not easy to know what, specifically, to listen for in the identification of Asian American voices. Many respondents based their judgments on intuition or instinct. One participant wrote, "I'm not really sure how to describe it... you just 'know' what it sounds like". Another: "I compare the voices that I am hearing to the past experiences where I have heard similar accents, and if I know which specific ethnicity the speaker is, I can use these experiences to match it in the future". Both of these speakers were White Americans who reported no specific linguistic experience with Korean. This feeling of intuition was echoed by one Korean American listener, who wrote, "I guess I think someone is Korean if they sound like any of the people who I grew up with. Honestly, I wouldn't be surprised if every person in the sound samples were all Korean people from California". This listener happened to be correct.

## 4. Discussion

The four listener groups perceived the same 39 voices differently overall, with subtle differences between Koreans, other Asians, and non-Asians. Korean listeners were the most likely to identify the voices as belonging to Asian Americans, and specifically, as belonging to Korean Americans. Non-Korean Asian listeners were somewhat less likely to identify the voices as belonging to Asians, but were much less likely to identify the voices as being Korean. Non-Asian People of Color and White listeners both identified the majority of the voices as belonging to White speakers and did not significantly differ from each other. Demographic differences between the speakers, such as their age of immigration or their generational status, did not affect their ratings of ethnic identification, but acoustic properties such as vocal pitch did correlate with ethnic identification.

If we assume that ethnically Korean listeners have, overall, the most experience listening to Korean voices, that non-Asian listeners have the least experience, and that non-Korean Asian listeners fall somewhere in between, then we could conclude that greater experience hearing Korean American voices leads to greater accuracy in identifying Korean ethnicity in unfamiliar voices. These results accord with similar findings in Hanna (1997), Newman and Wu (2011), and Wong and Babel (2017). Although it is of course possible for a non-Korean listener to have extensive experience with Korean voices, or for a Korean listener not to have heard many Korean voices in their life (e.g., transnational Korean adoptees raised in a predominantly non-Korean neighborhood), we believe in the safety

of this assumption: out of 105 listeners, 86 described themselves as something other than "Asian" for their own ethnic self-identification. Of these, 80 listeners (93%) indicated no experience with speaking Korean, and 76 (88%) indicated no experience with listening to Korean. In comparison, only one out of 19 Korean listeners indicated having no speaking and listening experience with Korean.

Experience with the Korean language is plainly not the same thing as experience of hearing Korean Americans speak English, but we consider the link between ethnicity, Korean language experience, and exposure to the Korean American ethnolect to be a strong one. (That said, a more detailed look at individual differences within each group, such as the responses of one White male listener who indicated an advanced level of experience with both speaking and reading Korean, would likely provide useful insights.)

Another way to interpret the results is through the concept of White normativity in the United States context. Devos and Banaji (2005), for example, found that White people and Asian people in the United States tend to associate the category of "American" with the White race (or the social construct of Whiteness). The strong associations in our data of American-born status and perceived Asianness and Whiteness are an example of this. Non-Asian POC and White listeners had a somewhat stronger negative association between Americanness and Asianness than Asian listeners, but the relationship existed for all listener groups.

It seems that all listeners, but especially White listeners, may have had the expectation that Whiteness is a kind of "default" in the context of speech, scientific experimentation, or both. In fact, the "genderless and raceless" problem in social science research has been raised often in recent years (Cundiff 2012; Salter and Adams 2013). Thus, barring any obviously perceptible accent (subjectively speaking, of course), White listeners would assume that the speaker is also White, as well as American. Asian listeners, on the other hand, are less likely to assume a default of Whiteness, which is why their scores for Whiteness likelihood are centered around "3" (Figure 5), while White listeners' scores center closer to "4".

We acknowledge the potential counterargument that people are simply more likely to think that an anonymized voice may be similar to themselves, rather than defaulting to White (cf. the auditory Own-Race Bias (Meissner and Brigham 2001; Perrachione et al. 2010)). Indeed, Korean listeners were most likely to select "Korean" as the specific Asian ethnicity when prompted, and Chinese or Taiwanese listeners were more likely to select "Chinese or Taiwanese", than Korean. However, the "like hears like" pattern did not extend to Vietnamese listeners, who were not more likely to identify the voices as Vietnamese American, nor to non-Asian POC listeners (comprising Black, Hispanic/Latino, and non-Asian mixed race listeners), who patterned almost identically to the White listener group in every result.

Whether being a member of an ethnic minority in a White-majority country intrinsically provides an individual with a greater diversity of experiences that affects their perception of ethnicity in speech or not, it seems that Asian listeners are less likely to set a default expectation of Whiteness when performing identification tasks that require sociocultural evaluation.

Some more evidence for this explanation comes from the post-hoc control study performed on twenty additional participants who were asked to read the transcripts of the audio clips instead of listening to them. In this case, eighteen of the readers identified as White, one as Hispanic/Latino, and one as Korean American. Results from the control study showed that even without any acoustic information, the non-Asian readers tended to score the speakers as likely to be White and not likely to be Asian. (Only one participant in the control study identified as Asian, and although their results constitute only one data point, they rated nearly every speaker as neutral.)

Taking into account that for both the main study and the control study, participants were not told that they would be presented with the speech of Korean Americans, and that for the control study, most participants still rated the speech transcripts as being more likely

to be from a White speaker than a speaker of other racial/ethnic groups, it is apparent that the influence of White normativity remains a very relevant factor in listener evaluations of ethnic identity.

We must also acknowledge that the design of this experiment likely played a role in the distribution of responses. Flege and Fletcher (1992) showed that increasing the range of variation in stimuli (in their case, including native English speakers along with Spanish and Chinese-accented English speakers in a foreign accent perception study) dramatically changed how listeners rated the strength of an accent. Increasing the range of stimuli gave listeners a different "baseline" of comparison, or calibration. We thus might expect that including White native speakers of American English in the stimuli, as past studies of Asian American English perception have done, could have resulted in more of the Korean speakers being rated as more likely to be Asian and Korean than White. On the other hand, it is equally plausible that we could have found no significant difference between White speakers and Korean speakers. This would be a fruitful avenue for follow-up research.

Drager (2010) wrote that individuals perceive speech differently depending on their own idiolect, their own experiences with other dialects, and their own attitudes toward the speaker (while Kang and Rubin (2009) also add that a listener's attitudes toward a speaker influences their perception of speech). The main effect of the perception study demonstrates how true this is, with a speaker's own ethnicity (as a proxy for their idiolect and dialect experience, but not for language attitudes) affecting their ability to identify Korean American speakers. The metalinguistic commentary given by the listeners also confirms this, with many non-Korean listeners describing their thought process for racial identification as matching the overall acoustic profile of the stimulus they heard with past experiences hearing Asian Americans speak, and one Korean listener explicitly stating that successful Korean identification meant directly comparing stimuli with the other Koreans in their life.

Without specific and sustained experience with Korean American speech in a listener's life, they are not likely to identify a voice as Korean American. The voice may sound "vaguely Asian", or perceptibly different from a White norm, but does not coalesce around a specifically "Korean" category for the purposes of accurate identification. Korean Americans with the requisite experience, however, were more certain that when one of the voices in this experiment was Asian, it was specifically a Korean voice (as opposed to a Chinese/Taiwanese voice or "other Asian").

The ability of Korean Americans to identify other Korean American voices may also be put in conversation with Levon (2014)'s exploration of the development and enregisterment of linguistic stereotypes. Specifically, several Korean American listeners cited the "Valley Girl" stereotype or persona when describing the speech of (female) Asian Americans in their metalinguistic commentary or folk linguistic narratives. Levon defined stereotype as "a cognitive structure that links group concepts with collections of trait attributes and roles" (2014, p. 544). In this case, some Korean Americans associate the group concept of "Korean" with the persona of "Valley Girl" and its social and linguistic traits: high rising terminals (uptalk), high pitch, use of salient lexical items such as "like", and being young, female, and from Southern California.

It is somewhat surprising that it was specifically the Korean American listeners, rather than the White listeners, who referenced an activated link between (some of) the variables that index Asian American and (some of) the variables that index Valley Girl. In the United States, the popular conceptualization of the Valley Girl is a White American woman, not a Korean American woman, but that could change over time within certain networks: speakers 08 and 30, mentioned earlier, fit the acoustic profile of the Valley Girl in their speech samples, and both were rated highly on the Asian likelihood scale and the American-born scale by Korean listeners. Some Asian American groups have been shown to participate strongly in, or even lead, sound changes in progress (Hall-Lew 2010); we believe the same is possible with listener-led development of new linguistic stereotypes.

Finally, we acknowledge that our perception experiment was not a test of a specific sociophonetic variable, and thus, does not offer any more definitive an answer to "What does a Korean/Asian American voice sound like?" than previous studies. However, the metalinguistic commentary offered by listeners does lay the groundwork for a host of potential future studies that examine segmental, prosodic, and lexical cues that index Korean/Asian American identity. Of particular interest is the role of prosody in dialect and ethnolect identification (Burdin et al. 2018), and our immediate future work will test for unique prosodic patterns in the voice samples beyond simple measures of mean pitch and pitch range. We hope to put this in conversation with sociocultural linguistic work that examines the construction of Asian American identity vis-à-vis White and Black identities and linguistic resources (cf. Bucholtz 2004; Chun 2001; Ito 2021).

## 5. Conclusions

We believe, as Campbell-Kibler (2010) writes, that perception is "at the heart" of variationist sociolinguistic research, and so, we conducted an experimental study of the perception of Korean American voices by Koreans and non-Koreans.

The first hypothesis—that listeners can correctly identify certain English-speaking voices as sounding distinctly Korean American—was proven true. This was dependent on the ethnicity of the listener, which in turn, proved hypothesis 1a: Korean listeners were the best at identifying the speakers as Korean American, while non-Korean Asian listeners were less accurate and non-Asian listeners were the least accurate. It was deemed plausible, though not entirely proven true, that listener familiarity with Korean American culture or Korean American voices would affect accuracy.

The second hypothesis—that identification as sounding Korean American would correlate with sounding non-native—was also proven true, albeit again dependent on the ethnicity of the listener: it was slightly more true for non-Asian listeners than for Asian listeners.

These data suggest that certain native English-speaking voices are indeed marked as sounding specifically Korean American, and consequently, that Korean Americans speaking English use some variation on a segmental and/or suprasegmental level to index their ethnic identity. The exact nature of this variation is beyond the scope of the study, but the preliminary analysis of pitch, as well as the metalinguistic commentary and folk linguistic narratives given by listeners, provides good indications of where to begin more focused experimental sociophonetic studies.

This work provides further evidence that the perception of social identities in speech is dependent on both speaker and listener characteristics, and also lays the groundwork for further exploration of the traits that make some Korean American and Asian American voices recognizable. Such traits may include ethnic rootedness and ethnic orientation among Korean Americans (Hoffman and Walker 2010; Reed 2018) or intersections of Korean (American) identity with other cultural identities (Choi 2015; Chun 2001), and linguists should also explore the role of bilingualism and bilingual phonology (Kang and Guion 2006) in ethnolect development. Finally, we hope that the discussion of ethnic identification in the context of the demonstration of White normativity in this work also highlights the need for a critical race sociolinguistics, echoing the calls to action sounded in this field and in others (Rosa and Flores 2017; Salter and Adams 2013).

**Author Contributions:** Conceptualization, A.C.; methodology, A.C. and S.C.; formal analysis, A.C.; writing—original draft preparation, A.C. and S.C.; writing–review and editing, A.C. and S.C.; visualization, A.C. All authors have read and agreed to the published version of the manuscript.

**Funding:** This research received no external funding.

**Informed Consent Statement:** Informed consent was obtained from all subjects involved in the study.

**Data Availability Statement:** Data will be publicly available on github at https://github.com/andrewqcheng, accessed on 2 November 2021.

**Acknowledgments:** The authors would like to thank audiences at the Linguistic Society of America and the University of California, Berkeley, as well as many anonymous reviewers, for their comments and feedback. Additional thanks to Sean Freeder for technical assistance with Amazon Mechanical Turk and Mark Chin for feedback on statistical testing. Finally, they express gratitude to the members of the Korean American community who lent their voices to this project. All errors are our own.

**Conflicts of Interest:** The authors declare no conflict of interest.

## Appendix A. Audio Stimulus Transcriptions

Audio stimulus transcriptions in conversation analysis style. Each speaker had two audio samples, identified with the speaker number and a random letter (A–E). The first sample was used in block A (race/ethnicity rating) and the second sample was used in block B (personality/birth status rating). The length of the clip in seconds is provided at the end of each transcription.

01A. They're little bit more tougher (.) a little bit more rougher (.) 'cause I think they've had a harder time (0:04)
01C. >Just on average yeah I would watch like< (1.5) like everyday actually (0:04)

02A. I don't know how they divided up but (1.2) I- I would (0:04)
02B. A lot of them know each otherˆ (.4) and because of that like (0:03)

03B. Like after a long time I would like >pick up my phone and talk to my mom< (0:03)
03C. Honestly I'm (.3) I think I'm content with where I amˆ (0:03)

04A. Someone upstairs yells and (.) some (0:02)
04C. But I definitely- (.4) do that 'cause I want him to- (.3) help me (0:04)

05A. I doˆ but I don't really listen to a lot of music in general (0:03)
05B. Hm that's a good question (1.0) Well there is a websiteˆ (0:04)

06B. So once they started popping up then we started eating out (0:03)
06D. If they're there (.7) but I don't necessarily seek them out (0:03)

07B. When I'm there (.2) I feel really comfortable but at the same time there's like a bit of like (0:04)
07D. I don't think I (.) purposely try to >but I just< (0:02)

08A. Yeah I thought it wasˆ (.2) pretty okay u:m (0:02)
08C. It's hard to: identify and like- (.6) see similarities (0:04)

09D. But I can't really gain any insight (0:02)
09E. Not- (.) like- (.) aggressively but if i can (0:02)

10B. Ve:ry (1.0) occasionally (0:03)
10C. So it switched (0.2) at some point (0:01)

11A. I- (0.5) think it's (0.3) pretty important (0:02)
11C. Depending on how the waiter looks (0:01)

12B. I think I (0.2) put more effort into learning it recently (0:02)
12D. I think (0.7) you can't fully identify with it (0:03)

13B. I do: (.) yeah (.) but something that's not as (0:02)
13D. Sometimes (0.4) whenever: (.) someone had a birthday (0:02)

14A. When I'm getting emotional or (0:01)
14B. Most like myself when I'm free: to switch (0:02)

15B. And I lived in like the center of (0:01)
15E. I think so: so I met some (0:02)

16B. I think it wasˆ (.) pretty good. 'cause I just talked to my mom (0:02)
16C. Back then my brother (0.3) he's five years younger than me so he was a little kid (0:03)

17A. And since it's just (0.35) me (0:02)
17C. Yeah it has happened (0.3) uh- it definitely wasn't (0.2) common (0:04)

19A. I have no idea (0:01)
19D. Yeah the thing is >that like< (.) >I think I was like pretty< (0.2) like on and off (0:02)

20C. I- yeah sometimes like when I- when I'm like surprisedˆ (0:02)
20D. I do think (.) it is important (0:02)

21B. U:m I remember that and the:n (0.3) sometimes my: friends would (0:03)
21D. So like when I'm really ti(h)red and frustrated sometimes I'll like (0:02)

22A. Uh that transition probably I: I don't (.) remember exact amount (0:03)
22B. Haven't really (.) thought about (0:02)

23B. Like my da:d doesn't make comments like that (0:02)
23C. Yaknow you're gonna (.) totally regret it u:m 'cause everyone else is (0:03)

24B. I- I have very vague memories of preschool but I remember I was (0:02)
24D. Being in conversation with them could be (0.4) .hh kinda hard sometimes (0:03)

25B. And he was like (.) you know doing whatever he was doing (0:02)
25C. Up until like six months ago I used to (h) watch a lot (0:02)

26B. The people he was currently working withˆ (0:01)
26C. I- I guess superficially I can say (0:02)

27A. Uh I think a lot of (.) I think I gained more confidence too (0:03)
27B. Yeah I think that was a surprising thing (0:02)

28C. I: (.5) think I (.) enjoyed it for the most partˆ (0:03)
28D. Yeah there's a big difference I think (.) especially (0:02)

29A. I mean >I remember like hanging out with kids there< and (.4) I don- I never had trouble (0:03)
29C. There- there was a lot of pressure on me >but now it's just kinda like< (0:02)

30A. You knowˆ (.) I'd say it comes back (.3) fairly quickly (0:03)
30C. I think that no::w they care a lot because (0:03)

31C. It was different when I just went off and did my own thing (0:02)
31D. Maybe this is true (.) like maybe this is actually happening (0:03)

32A. Like let's say a situation has happenedˆ and we're both trying to explainˆ (0:03)
32B. Yo::uˆ can make your own choicesˆ and yo:uˆ can be independentˆ (0:04)

33A. It would be: if I: were going to the market or if I were going to a restaurant (0:04)
33B. I remember specifically: (0.2) uh thinking (0:03)

34A. So it's a such a valuable to:ol that I don't (0:02)
34C. But you know in retrospect I think that's what helped (.) me and my sister (0:03)

35A. I (.) do have va:gue memories of being set- like pulled aside for (0:04)
35B. I mean it's- it's- it'll take me where I need to go (0:02)

36A. I'm comfortable as long as it doesn't get (.) too complicated (0:03)
36C. I like whatever's on the radio whatever hits there are but then um (0:04)

37B. I think you can get away with it as long as they're youngˆ (0:02)
37C. Uh and even my own family they only lived there fo:r (0:02)

38A. Yeah that would be: I would consider that to be like (0:03)
38C. Or I feel like it's more apparent with (0:02)

39C. ˆSometimes we switch (.) but that's for like (0:03)
39D. So that was a little different from what I expectedˆ (0:02)

40C. And nowˆ it's grown so much (0:01)
40D. We:ll you know whatˆ (.) um they di:d um much later (0:03)

## Appendix B. Speaker Ratings by Listener Ethnic Group

Each speaker was rated on Likert scales of likelihood of being White, Asian, and American-born, by listeners who were Korean, non-Korean Asians, non-Asian People of Color, or White. Mean scores per listener group are reported. "Cal." refers to whether the speaker grew up in California or not. "KAS" refers to a 10-point scale of "Korean cultural adjacency", for which higher scores reflect greater access to or experience with Korean language and culture in childhood and adulthood (Cheng 2020b).

| Subj. | Age | Gender | Cal. | AOA | Gen. | Dominant Lang. | KAS | Listener Ethnic Group | White | Asian | American |
|---|---|---|---|---|---|---|---|---|---|---|---|
| 01 | 20 | Male | Yes | 0 | 2nd | English | 3.1 | Asian_Korean_POC | 2.947368 | 3.789474 | 4.578947 |
| | | | | | | | | Asian_Not Korean_POC | 4.166667 | 2.666667 | 4.277778 |
| | | | | | | | | Not Asian_Not Korean_POC | 4.190476 | 2.47619 | 4.285714 |
| | | | | | | | | Not Asian_Not Korean_White | 4.361702 | 2.340426 | 4.297872 |
| 02 | 21 | Male | Yes | 9 | 1.5 | English | 6.35 | Asian_Korean_POC | 1.736842 | 4.578947 | 3.631579 |
| | | | | | | | | Asian_Not Korean_POC | 1.777778 | 4.5 | 3.166667 |
| | | | | | | | | Not Asian_Not Korean_POC | 2.761905 | 3.428571 | 3.761905 |
| | | | | | | | | Not Asian_Not Korean_White | 2.723404 | 3.276596 | 3.404255 |
| 03 | 19 | Female | Yes | 0 | 2nd | English | 4.1 | Asian_Korean_POC | 2.894737 | 4.315789 | 3.578947 |
| | | | | | | | | Asian_Not Korean_POC | 3.277778 | 3.833333 | 3.666667 |
| | | | | | | | | Not Asian_Not Korean_POC | 3.857143 | 2.52381 | 4.285714 |
| | | | | | | | | Not Asian_Not Korean_White | 3.808511 | 2.468085 | 4.12766 |
| 04 | 25 | Female | Yes | 0.5 | 1.5 | English | 2.7 | Asian_Korean_POC | 2.894737 | 3.894737 | 4.210526 |
| | | | | | | | | Asian_Not Korean_POC | 2.777778 | 3.5 | 4 |
| | | | | | | | | Not Asian_Not Korean_POC | 3.619048 | 2.52381 | 4.285714 |
| | | | | | | | | Not Asian_Not Korean_White | 4 | 1.978723 | 4.361702 |
| 05 | 20 | Female | Yes | 3 | 1.5 | English | 5.6 | Asian_Korean_POC | 2.368421 | 4.263158 | 2.157895 |
| | | | | | | | | Asian_Not Korean_POC | 2.555556 | 4.222222 | 2.611111 |
| | | | | | | | | Not Asian_Not Korean_POC | 3.380952 | 3.52381 | 2.761905 |
| | | | | | | | | Not Asian_Not Korean_White | 3.765957 | 3 | 3.234043 |
| 06 | 18 | Female | No | 5 | 1.5 | English | 4.3 | Asian_Korean_POC | 3.210526 | 3.526316 | 4.210526 |
| | | | | | | | | Asian_Not Korean_POC | 3.5 | 3.111111 | 4.388889 |
| | | | | | | | | Not Asian_Not Korean_POC | 3.285714 | 2.285714 | 4.380952 |
| | | | | | | | | Not Asian_Not Korean_White | 3.787234 | 2.297872 | 4 |
| 07 | 19 | Female | Yes | 8 | 1.5 | Both | 5.75 | Asian_Korean_POC | 2.315789 | 3.947368 | 1.894737 |
| | | | | | | | | Asian_Not Korean_POC | 2.055556 | 3.777778 | 2.444444 |
| | | | | | | | | Not Asian_Not Korean_POC | 2.666667 | 2.809524 | 3.190476 |
| | | | | | | | | Not Asian_Not Korean_White | 2.468085 | 2.468085 | 2.914894 |
| 08 | 20 | Female | Yes | 0 | 2nd | English | 5.7 | Asian_Korean_POC | 2.473684 | 4.421053 | 3.736842 |
| | | | | | | | | Asian_Not Korean_POC | 2.888889 | 4.111111 | 3.5 |
| | | | | | | | | Not Asian_Not Korean_POC | 3.761905 | 3.142857 | 4 |
| | | | | | | | | Not Asian_Not Korean_White | 3.638298 | 3 | 3.914894 |
| 09 | 20 | Male | Yes | 0 | 2nd | English | 5 | Asian_Korean_POC | 2.736842 | 3.947368 | 3.578947 |
| | | | | | | | | Asian_Not Korean_POC | 3.222222 | 3.388889 | 3.277778 |
| | | | | | | | | Not Asian_Not Korean_POC | 3.809524 | 2.952381 | 3.761905 |
| | | | | | | | | Not Asian_Not Korean_White | 3.148936 | 2.851064 | 3.340426 |
| 10 | 22 | Female | Yes | 0 | 2nd | English | 5.35 | Asian_Korean_POC | 2.736842 | 3.894737 | 3 |
| | | | | | | | | Asian_Not Korean_POC | 3.111111 | 3.722222 | 2.833333 |
| | | | | | | | | Not Asian_Not Korean_POC | 3.761905 | 2.809524 | 3.142857 |
| | | | | | | | | Not Asian_Not Korean_White | 4.042553 | 2.382979 | 2.957447 |
| 11 | 21 | Male | Yes | 10 | 1.5 | English | 6.5 | Asian_Korean_POC | 2.894737 | 4.052632 | 3.736842 |
| | | | | | | | | Asian_Not Korean_POC | 3.166667 | 3.777778 | 3.722222 |
| | | | | | | | | Not Asian_Not Korean_POC | 3.857143 | 3.095238 | 3.809524 |
| | | | | | | | | Not Asian_Not Korean_White | 3.382979 | 3.06383 | 3.978723 |

| Subj. | Age | Gender | Cal. | AOA | Gen. | Dominant Lang. | KAS | Listener Ethnic Group | White | Asian | American |
|---|---|---|---|---|---|---|---|---|---|---|---|
| 12 | 23 | Male | No | 0 | 2nd | English | 5.55 | Asian_Korean_POC | 2.263158 | 4.263158 | 2.368421 |
| | | | | | | | | Asian_Not Korean_POC | 2.277778 | 4.111111 | 2.833333 |
| | | | | | | | | Not Asian_Not Korean_POC | 2.761905 | 3.238095 | 2.333333 |
| | | | | | | | | Not Asian_Not Korean_White | 2.702128 | 3.319149 | 2.234043 |
| 13 | 20 | Female | Yes | 0 | 2nd | English | 3.85 | Asian_Korean_POC | 2.631579 | 3.894737 | 3.157895 |
| | | | | | | | | Asian_Not Korean_POC | 3.333333 | 3.722222 | 3.444444 |
| | | | | | | | | Not Asian_Not Korean_POC | 3.619048 | 2.952381 | 3.238095 |
| | | | | | | | | Not Asian_Not Korean_White | 3.893617 | 2.765957 | 3.468085 |
| 14 | 25 | Female | Yes | 0 | 2nd | English | 6.5 | Asian_Korean_POC | 3.421053 | 3.263158 | 3.736842 |
| | | | | | | | | Asian_Not Korean_POC | 3.388889 | 3.222222 | 3.222222 |
| | | | | | | | | Not Asian_Not Korean_POC | 3.666667 | 2.47619 | 3.47619 |
| | | | | | | | | Not Asian_Not Korean_White | 3.978723 | 2.212766 | 3.319149 |
| 15 | 26 | Female | No | 10 | 1.5 | Both | 4.6 | Asian_Korean_POC | 2.473684 | 3.894737 | 3.052632 |
| | | | | | | | | Asian_Not Korean_POC | 3.388889 | 3.333333 | 3.111111 |
| | | | | | | | | Not Asian_Not Korean_POC | 3.666667 | 2.857143 | 3.285714 |
| | | | | | | | | Not Asian_Not Korean_White | 3.638298 | 2.702128 | 2.680851 |
| 16 | 19 | Female | No | 3 | 1.5 | English | 4.8 | Asian_Korean_POC | 2.736842 | 4 | 2.736842 |
| | | | | | | | | Asian_Not Korean_POC | 3.111111 | 3.944444 | 2.611111 |
| | | | | | | | | Not Asian_Not Korean_POC | 4.047619 | 3 | 3.190476 |
| | | | | | | | | Not Asian_Not Korean_White | 3.829787 | 2.638298 | 3.404255 |
| 17 | 18 | Female | No | 10 | 1.5 | Both | 6.75 | Asian_Korean_POC | 3.157895 | 3.473684 | 3.263158 |
| | | | | | | | | Asian_Not Korean_POC | 3 | 3.444444 | 2.888889 |
| | | | | | | | | Not Asian_Not Korean_POC | 3.666667 | 2.52381 | 3.952381 |
| | | | | | | | | Not Asian_Not Korean_White | 3.829787 | 2.744681 | 3.042553 |
| 19 | 20 | Male | Yes | 0 | 2nd | English | 3.85 | Asian_Korean_POC | 3.315789 | 3.526316 | 3.736842 |
| | | | | | | | | Asian_Not Korean_POC | 3.5 | 3.5 | 3.333333 |
| | | | | | | | | Not Asian_Not Korean_POC | 3.714286 | 3.047619 | 3.190476 |
| | | | | | | | | Not Asian_Not Korean_White | 4.042553 | 2.595745 | 3.234043 |
| 20 | 19 | Female | No | 0 | 2nd | English | 5.35 | Asian_Korean_POC | 2.052632 | 4.473684 | 2.736842 |
| | | | | | | | | Asian_Not Korean_POC | 2.166667 | 4.333333 | 3.166667 |
| | | | | | | | | Not Asian_Not Korean_POC | 2.619048 | 3.380952 | 2.809524 |
| | | | | | | | | Not Asian_Not Korean_White | 2.808511 | 3.404255 | 2.659574 |
| 21 | 27 | Female | No | 0 | 2nd | English | 2.8 | Asian_Korean_POC | 3.368421 | 3.789474 | 4.368421 |
| | | | | | | | | Asian_Not Korean_POC | 3.833333 | 3.444444 | 4.333333 |
| | | | | | | | | Not Asian_Not Korean_POC | 4.380952 | 2.238095 | 4.761905 |
| | | | | | | | | Not Asian_Not Korean_White | 4.340426 | 2.340426 | 4.446809 |
| 22 | 29 | Male | Yes | 0 | 2nd | English | 3.8 | Asian_Korean_POC | 2 | 4.210526 | 3.105263 |
| | | | | | | | | Asian_Not Korean_POC | 2.222222 | 4.388889 | 3.166667 |
| | | | | | | | | Not Asian_Not Korean_POC | 3.285714 | 3.428571 | 3.333333 |
| | | | | | | | | Not Asian_Not Korean_White | 3.12766 | 3.468085 | 3.148936 |
| 23 | 28 | Female | Yes | 0 | 2nd | English | 7.3 | Asian_Korean_POC | 3.210526 | 4.052632 | 4.473684 |
| | | | | | | | | Asian_Not Korean_POC | 3.555556 | 3.666667 | 4.333333 |
| | | | | | | | | Not Asian_Not Korean_POC | 4.380952 | 2.238095 | 4.761905 |
| | | | | | | | | Not Asian_Not Korean_White | 4.276596 | 2.340426 | 4.553191 |
| 24 | 18 | Male | Yes | 0 | 2nd | English | 3.75 | Asian_Korean_POC | 2.315789 | 4.263158 | 3.105263 |
| | | | | | | | | Asian_Not Korean_POC | 2.666667 | 4.277778 | 3.388889 |
| | | | | | | | | Not Asian_Not Korean_POC | 3.333333 | 3.142857 | 3.428571 |
| | | | | | | | | Not Asian_Not Korean_White | 3.255319 | 3.510638 | 3.595745 |
| 25 | 24 | Female | Yes | 0 | 2nd | Both | 7.15 | Asian_Korean_POC | 1.947368 | 4.473684 | 1.947368 |
| | | | | | | | | Asian_Not Korean_POC | 2.444444 | 3.444444 | 1.833333 |
| | | | | | | | | Not Asian_Not Korean_POC | 2.714286 | 2.714286 | 1.714286 |
| | | | | | | | | Not Asian_Not Korean_White | 3.12766 | 2.87234 | 1.574468 |
| 26 | 26 | Male | Yes | 0 | 2nd | English | 5.6 | Asian_Korean_POC | 2.842105 | 4.105263 | 3.578947 |
| | | | | | | | | Asian_Not Korean_POC | 3.277778 | 3.611111 | 3.777778 |
| | | | | | | | | Not Asian_Not Korean_POC | 3.952381 | 2.904762 | 4.238095 |
| | | | | | | | | Not Asian_Not Korean_White | 3.87234 | 2.851064 | 4.106383 |
| 27 | 23 | Male | Yes | 12 | 1.5 | English | 5.4 | Asian_Korean_POC | 2.315789 | 3.842105 | 2.368421 |
| | | | | | | | | Asian_Not Korean_POC | 2.666667 | 3.611111 | 2.888889 |
| | | | | | | | | Not Asian_Not Korean_POC | 3.666667 | 2.761905 | 3.380952 |
| | | | | | | | | Not Asian_Not Korean_White | 3.595745 | 2.553191 | 3.489362 |
| 28 | 21 | Female | Yes | 0 | 2nd | English | 4 | Asian_Korean_POC | 2.473684 | 4.105263 | 3.684211 |
| | | | | | | | | Asian_Not Korean_POC | 3.5 | 3.055556 | 3.888889 |
| | | | | | | | | Not Asian_Not Korean_POC | 3.47619 | 2.809524 | 3.857143 |
| | | | | | | | | Not Asian_Not Korean_White | 3.744681 | 2.617021 | 3.595745 |

| Subj. | Age | Gender | Cal. | AOA | Gen. | Dominant Lang. | KAS | Listener Ethnic Group | White | Asian | American |
|---|---|---|---|---|---|---|---|---|---|---|---|
| 29 | 25 | Male | Yes | 0 | 2nd | English | 4.65 | Asian_Korean_POC | 2.842105 | 3.894737 | 3.789474 |
| | | | | | | | | Asian_Not Korean_POC | 3.222222 | 3.944444 | 3.611111 |
| | | | | | | | | Not Asian_Not Korean_POC | 3.619048 | 2.47619 | 4.238095 |
| | | | | | | | | Not Asian_Not Korean_White | 3.93617 | 2.617021 | 3.744681 |
| 30 | 30 | Female | Yes | 0 | 2nd | English | 6.56 | Asian_Korean_POC | 2.631579 | 4.105263 | 4.368421 |
| | | | | | | | | Asian_Not Korean_POC | 2.944444 | 3.555556 | 4.333333 |
| | | | | | | | | Not Asian_Not Korean_POC | 3.809524 | 2.571429 | 4.761905 |
| | | | | | | | | Not Asian_Not Korean_White | 3.702128 | 2.638298 | 4.06383 |
| 31 | 25 | Male | Yes | 0 | 2nd | English | 6.2 | Asian_Korean_POC | 2.526316 | 3.526316 | 4.368421 |
| | | | | | | | | Asian_Not Korean_POC | 3.388889 | 3.388889 | 3.666667 |
| | | | | | | | | Not Asian_Not Korean_POC | 4.095238 | 2.571429 | 4.142857 |
| | | | | | | | | Not Asian_Not Korean_White | 3.765957 | 2.617021 | 3.93617 |
| 32 | 26 | Male | Yes | 0 | 2nd | English | 3.65 | Asian_Korean_POC | 2.157895 | 4.263158 | 3.263158 |
| | | | | | | | | Asian_Not Korean_POC | 2.444444 | 3.777778 | 3.055556 |
| | | | | | | | | Not Asian_Not Korean_POC | 3.142857 | 3 | 3.47619 |
| | | | | | | | | Not Asian_Not Korean_White | 2.744681 | 2.93617 | 3.191489 |
| 33 | 30 | Female | No | 3 | 1.5 | English | 5.6 | Asian_Korean_POC | 3.368421 | 3.947368 | 3.210526 |
| | | | | | | | | Asian_Not Korean_POC | 2.888889 | 3.555556 | 3.166667 |
| | | | | | | | | Not Asian_Not Korean_POC | 3.761905 | 2.714286 | 3.52381 |
| | | | | | | | | Not Asian_Not Korean_White | 3.808511 | 2.638298 | 3.06383 |
| 34 | 32 | Female | Yes | 10 | 1.5 | English | 5.15 | Asian_Korean_POC | 2.842105 | 3.947368 | 3.473684 |
| | | | | | | | | Asian_Not Korean_POC | 3.166667 | 3.833333 | 3.944444 |
| | | | | | | | | Not Asian_Not Korean_POC | 4.285714 | 2.52381 | 4.428571 |
| | | | | | | | | Not Asian_Not Korean_White | 4.212766 | 2.361702 | 3.808511 |
| 35 | 28 | Female | Yes | 0 | 2nd | English | 4.6 | Asian_Korean_POC | 3.684211 | 3.789474 | 4.105263 |
| | | | | | | | | Asian_Not Korean_POC | 4 | 3.111111 | 4.666667 |
| | | | | | | | | Not Asian_Not Korean_POC | 4.428571 | 2.47619 | 4.380952 |
| | | | | | | | | Not Asian_Not Korean_White | 4.319149 | 2.191489 | 4.297872 |
| 36 | 31 | Female | Yes | 0 | 2nd | English | 5.15 | Asian_Korean_POC | 2.052632 | 4.315789 | 2.842105 |
| | | | | | | | | Asian_Not Korean_POC | 2.111111 | 4.5 | 3.055556 |
| | | | | | | | | Not Asian_Not Korean_POC | 3.47619 | 3.142857 | 3.47619 |
| | | | | | | | | Not Asian_Not Korean_White | 3.617021 | 3.106383 | 3.276596 |
| 37 | 36 | Female | Yes | 0 | 2nd | English | 6.25 | Asian_Korean_POC | 3.631579 | 3.315789 | 3.473684 |
| | | | | | | | | Asian_Not Korean_POC | 3.666667 | 2.611111 | 3.611111 |
| | | | | | | | | Not Asian_Not Korean_POC | 3.380952 | 2.190476 | 3.380952 |
| | | | | | | | | Not Asian_Not Korean_White | 3.680851 | 1.978723 | 3.510638 |
| 38 | 25 | Male | Yes | 0 | 2nd | English | 4.4 | Asian_Korean_POC | 3.210526 | 3.631579 | 3.368421 |
| | | | | | | | | Asian_Not Korean_POC | 3.944444 | 2.944444 | 3.777778 |
| | | | | | | | | Not Asian_Not Korean_POC | 3.952381 | 2.428571 | 3.666667 |
| | | | | | | | | Not Asian_Not Korean_White | 4.12766 | 2.191489 | 3.851064 |
| 39 | 32 | Female | Yes | 16 | 1.5 | Both | 5.5 | Asian_Korean_POC | 1.947368 | 4.578947 | 2.736842 |
| | | | | | | | | Asian_Not Korean_POC | 1.944444 | 4.277778 | 3.277778 |
| | | | | | | | | Not Asian_Not Korean_POC | 2.619048 | 4.190476 | 3.666667 |
| | | | | | | | | Not Asian_Not Korean_White | 2.468085 | 3.87234 | 3.297872 |
| 40 | 55 | Female | Yes | 8 | 1.5 | Both | 5.15 | Asian_Korean_POC | 3.368421 | 3.578947 | 3.263158 |
| | | | | | | | | Asian_Not Korean_POC | 3.388889 | 3.277778 | 3.222222 |
| | | | | | | | | Not Asian_Not Korean_POC | 4.095238 | 2.333333 | 3.285714 |
| | | | | | | | | Not Asian_Not Korean_White | 3.957447 | 2.042553 | 3.489362 |

## Notes

[1] However, it is often construed as a fundamental pillar of the uniquely U.S. American "ethnoracial pentagon" (Torres-Saillant 2003): White, Black, Asian, Hispanic/Latino, and Native American/Indigenous, including Alaska Native and Native Hawaiian.

[2] With several exceptions, including Hmong and Kurdish.

[3] Note that it is also common for out-group members to have different associations between signals and categories compared to in-group members, rather than to have no associations at all, as has been shown in Johnstone and Kiesling (2008) and Villarreal (2016), inter alia.

[4] Despite the variation, any audio sampled at these rates and a bit depth of 16 is considered standard for most speech perception experiments.

[5] Thanks to an anonymous reviewer for noting the potential bias that could be introduced this way: due to the task design, over time, participants may have begun to intuit that the experiment was specifically targeting perception of Asian American voices. Great care was taken to ensure participant naivety to the goal of the task, but the possibility of this intuition always remains.

6　　To account for potential variability in the 1.5 group due to the wide range of ages of arrival, the same tests were run with the speakers separated into three groups: second generation, "early" 1.5-generation arrivals (who immigrated prior to age 6) and "late" 1.5-generation arrivals (who immigrated between 6 and 16). The differences were similarly insignificant.

7　　This category included those who identified as Hispanic/Latino, African American, Indigenous, and non-Asian mixed race.

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
