# Peer review of "The Effect of Ethnicity on Identification of Korean American Speech"

_languages, doi:10.3390/languages6040186_

Round 1

Reviewer 1 Report

The present paper reports on an ethnolect identification study, focusing on Korean American speech. I commend the author(s) for an interesting paper with intriguing results. I believe that this paper has potential, but I think more revision is necessary. Thus, I recommend ‘revise and resubmit’. Below I outline some issues that I believe would need to be addressed.

First and foremost, the author(s) discuss on page 11 that the Korean American listeners were ‘the most accurate in the identification of the speakers’ (line 395). However, since all the speakers were Korean American, any identification of the voices as Korean would be correct. There also appears to be a tendency for groups to select their own group as the response. In the paragraph on pg. 11 starting with line 382, the authors describe that Chinese and Taiwanese listeners tended to select Chinese or Taiwanese as their response (along with the tendency for White listeners to select the voices as White). So, perhaps this result may be driven more by choosing one’s own group than hearing a separate variety, as an audio version of the Own-Race Bias. Additionally, the Vietnamese listeners were as accurate as the Korean listeners. So, I am somewhat confused by the conclusion that the Korean listeners are the most accurate. One might better state that Vietnamese and Korean listeners are the most accurate.

To address these concerns, I would encourage the author(s) to perhaps dig deeper into the performance of the non-Korean Asian listeners and elaborate on these results. A fuller discussion, and perhaps a more detailed description of the results, would have the potential to alleviate this concern.

Second, while it is not the goal of the paper, one rather quick way to demonstrate that some of the results are not a result of some type of own-race bias would be to take a very quick look at one feature and correlate with the judgment. For example, since many of the qualitative responses mentioned pitch, if the samples that had a higher pitch were more likely to be called Korean, that would greatly strengthen the conclusions. This feature comparison does not have to be extensive, and since there are only 39 clips, getting an average pitch (or perhaps a measure of HRT) and then some type of correlation or simple regression could be added with a modicum of effort.

Third, more information regarding the recording situation and the respondent context is always helpful. How were the speakers recorded? What equipment was used? What microphones? What was the bit depth and sample rate? This could have an impact on how we interpret the results, as depending on the recordings, certain aspects of pitch, prosody, or voice quality (mentioned as possible features of Korean American English) are not captured with fidelity. Additionally, the paper describes that listeners used ‘personal audio devices on their own computers’ (line 222). What does this mean exactly? For example, does this mean headphones were required? Or could a listener use the in-built speakers of the device? Could they use a cell phone or tablet? All of these can have an impact as to how much detail the listener could realistically hear.

In sum, I find parts of the paper compelling, and there are the makings of a meaningful contribution. With these concerns addressed, the paper would be greatly strengthened, and I think would be a robust addition to the journal.

Author Response

Reviewer 1:

We have clarified the working definition of “accuracy” in section 3.3, as well as included a new figure that illustrates the differences between Asian ethnic groups in ethnic identification. It remains true that the Korean listeners, taken together, had the highest ratio for choosing “Korean” as the specific ethnicity, which is why we consider them to be the most accurate.

We have included a new figure that illustrates the described findings for the Asian listener groups in more detail.

Due to the time constraint, we have declined to do a correlation analysis of any one feature with the race ratings. However, we have included an expanded impressionistic analysis of the highest and lowest-rated voices for each listener group. In addition, we have plans to do a detailed acoustic analysis for a future paper.

We have added more detail about the sampling rate and bit depth of the audio (all of it was standard quality audio or higher). We have also added detail about the listeners’ audio equipment for listening (headphones, earbuds, or external speakers).

Thank you for your reviews!

Author Response

The writing style of the introduction has been amended to reduce the frequency of the word “they”.

We have a new paragraph focused on new dialect emergence and ethnic rootedness to the end of section 1.1, and better connect it to section 1.2.

We have added references to the Miami English studies to the paragraph about Chicano English.

The paragraph about growing variationist research has been clarified to refer to Asian American speech communities.

I have added the reference to Lee 2016 and expanded the discussion of the three cited articles on Korean American variation in English production.

We have removed the paragraph about perception of Korean American accents; on second read, it doesn’t make sense in the context of the section.

We have amended the phrasing of “more/less experience” to being a member of the in-group or out-group, as we did not actually test for the effects of experience on listener performance.

We have clarified the speaker population: “Samples of natural speech were taken from thirty-nine Korean Americans with two parents who were ethnic Koreans”

We have added to the Appendix a transcription of all 78 sentences used in the experiment. The transcription has been done according to conversation analysis methodology. We do not believe that any sub-group of the speakers was more likely to talk about one topic than another, since they were all asked the same set of questions in the semi-structured interview.

We have clarified that all of the speakers were ethnically Korean with two Korean parents; i.e., no speakers were multiracial. In addition, none of the speakers who identified as Asian were multiracial (which was unintentional, but did make it unnecessary to consider the interesting positionality of Asians who are also White/Black/etc.). We have clarified the number of second and 1.5 generation Korean Americans.

The Likert scales were adapted from Campbell-Kibler (2007); we have added the relevant citation.

We have clarified that the readers in the post-hoc test were not reading aloud/recording their own speech, but only silently reading the transcriptions of the audio. We have changed their description from “listeners” to “participants”, since they did not listen to anything. Also, we have clarified that “neutral” in this case refers to the Likert scale choice on the different race/ethnicity scales.

We have reconfigured the figures so that no graph is faceted by the speaker's generational status (Previously Figure 2; this is now Figure 4). We have also reconstructed them so that the “American-born” box is separate from the race/ethnicity boxes. Finally, we have clarified in the results section what “mean rating” means in a new paragraph that begins section 3.1.

We have added the number of Asian/Non-Asian (etc.) listeners to the captions of each figure.

“this chart shows why there is a problem with this study. Why isn't 'Black' following the 'White' category? I suspect there is a sampling problem and that your regression lines are fixed--if you used something like 'cubic' would you reach a different generalization? It looks like there is one Black person with 15+ years and this may mean African, not African-American. Yet you're treating them the same as the distribution in other groups. If you say they don't matter, then they should have been excluded (and noted that they were excluded in the methods section).”

We are unsure how to interpret this comment. There were no black speakers; In Figure 3, the “Black” line indicates listener perceptions of Blackness based on the (Korean) speakers’ age of arrival, and the line is flat all the way across at about 2-2.5 (on the y-axis). This indicates that regardless of AOA, all speakers were perceived as being unlikely to be Black.

We have added ellipses to the scatterplots and thank the reviewer for this great visualization suggestion.

As for the Kendall test, we specifically chose this test because it was parametric and the data didn’t pass the Shapiro test of normality. Kendall’s tau indicates more discordant pairs with a value closer to -1; in this data, the NKA group has the most discordant pairs and indeed has the tau value closest to -1 (at -0.62), compared to the other three groups.

We were advised by previous reviewers to look into the correlation specifically using a statistical test (rather than simply observing it, as we had done previously), and we do not think it is superfluous. We recognize that of course a speaker could be heard as (and also identify as) both White and Asian, but the overall trend does point to an apparent assumption on the part of listeners that the categories are (sometimes) exclusive.

We have clarified that the choices for listeners with respect to foreign-born status were “American-born” and “foreign-born”. Because this study took place in the United States with speakers and listeners from the United States, we did not consider Canada or the UK, and acknowledge that this is a potential area for future research.

We have added the focus on Korean Californians to the abstract and introduction, but we do not believe that the title needs to be changed. (We can leave the final decision up to the editor.)

We added a brief impressionistic acoustic analysis of some individual speakers, although we maintain that an in-depth acoustic analysis (with formant measurements, etc.) is beyond the scope of the current study and is in fact planned for a future study.

Thank you for your review!

Round 2

Reviewer 1 Report

 The present revised paper is an improvement. I believe that the authors have addressed most of my concerns. 

I must note, however, that I feel that my second recommendation, which was not included in this version of the paper, remains valid. While a full comparison of multiple phonetic features is beyond the scope of the present paper, a simple correlation between one feature, such as pitch, would strengthen the present version. In fact, I still encourage the authors to consider it, as the impact of the results could greatly enhance the overall work. One could simply get an F0 measure from the 39 clips (there are scripts for Praat or MatLab that can get this for you in seconds) and perform a simple correlation with the ethnicity judgements. Such an addition would not require more than a modicum of effort. 

Author Response

We have added the pitch correlation analysis. No correlations between mean pitch of utterance or pitch range of utterance and ethnicity ratings for male speakers. For female speakers, higher pitch correlated with higher Asian ratings and lower White ratings, but greater pitch range did not correlate as strongly. These results are reported in detail in the results section, along with a Figure demonstrating some of the correlations and a Table reporting some of the Kendall test results.

Reviewer 2 Report

Thank you for making the changes you did. The paper reads much better and will make an important contribution to the discussion on Asian Americans, but more specifically to continue work on Korean Americans.

Author Response

Thank you!